# Modulation of Phosphate Deficiency-Induced Metabolic Changes by Iron Availability in *Arabidopsis thaliana*

**DOI:** 10.3390/ijms22147609

**Published:** 2021-07-16

**Authors:** Ranju Chutia, Sarah Scharfenberg, Steffen Neumann, Steffen Abel, Jörg Ziegler

**Affiliations:** 1Department of Molecular Signal Processing, Leibniz Institute of Plant Biochemistry, Weinberg 3, D-06120 Halle, Germany; rchutia@mpipz.mpg.de (R.C.); steffen.abel@ipb-halle.de (S.A.); 2Synergy Research Group Bioinformatics & Scientific Data, Leibniz Institute of Plant Biochemistry, Weinberg 3, D-06120 Halle, Germany; sarah.scharfenberg@ipb-halle.de (S.S.); steffen.neumann@ipb-halle.de (S.N.); 3German Centre for Integrative Biodiversity Research (iDiv) Halle-Jena- Leipzig, Deutscher Platz 5e, D-04103 Leipzig, Germany; 4Department of Plant Science, University of California-Davis, Davis, CA 95616, USA

**Keywords:** combined nutrient deficiency, phosphate, iron, *pho1*, *Arabidopsis thaliana*, amino acids, organic acids

## Abstract

Concurrent suboptimal supply of several nutrients requires the coordination of nutrient-specific transcriptional, phenotypic, and metabolic changes in plants in order to optimize growth and development in most agricultural and natural ecosystems. Phosphate (P_i_) and iron (Fe) deficiency induce overlapping but mostly opposing transcriptional and root growth responses in *Arabidopsis thaliana*. On the metabolite level, P_i_ deficiency negatively modulates Fe deficiency-induced coumarin accumulation, which is controlled by Fe as well as P_i_ deficiency response regulators. Here, we report the impact of Fe availability on seedling growth under P_i_ limiting conditions and on P_i_ deficiency-induced accumulation of amino acids and organic acids, which play important roles in P_i_ use efficiency. Fe deficiency in P_i_ replete conditions hardly changed growth and metabolite profiles in roots and shoots of *Arabidopsis thaliana*, but partially rescued growth under conditions of P_i_ starvation and severely modulated P_i_ deficiency-induced metabolic adjustments. Analysis of T-DNA insertion lines revealed the concerted coordination of metabolic profiles by regulators of Fe (FIT, bHLH104, BRUTUS, PYE) as well as of P_i_ (SPX1, PHR1, PHL1, bHLH32) starvation responses. The results show the interdependency of P_i_ and Fe availability and the interplay between P_i_ and Fe starvation signaling on the generation of plant metabolite profiles.

## 1. Introduction

Plant growth and development strongly depends on the availability of nutrients in agricultural and natural ecosystems. The efficiency with which plants utilize a certain nutrient is determined by the supply of a well-balanced ratio of several nutrients. Therefore, an increasing number of studies investigate the coordination of plant responses to nutrient ratios in order to elucidate the molecular components, which ensures optimal provision with all required nutrients under conditions of changing nutrient supply [1]. For example, phosphate deficiency regulators were shown to play an important role in the integration of phosphate and nitrate nutritional signals [2,3]. Phosphate deficiency regulators are also involved in the control of homeostasis of other macronutrients and of metals, such as sulfur and zinc, respectively [4].

Phosphorus possesses crucial roles in energy and nucleotide metabolism and as a constituent of membranes. It is taken up by the plant in the form of inorganic phosphate (P_i_, H_2_PO_4_^−^ or HPO_4_^2−^) by epidermal and cortical cells of the roots and transferred to the xylem for translocation to the shoots, a process which is mediated by PHO 1 (PHOSPHATE1) in *Arabidopsis thaliana* [5,6,7]. P_i_ deficiency leads to profound phenotypic changes such as decreased shoot weight and increased root to shoot ratios, which are accompanied by reprogramming of cellular metabolism, such as the substitution of phospholipids by sulfolipids or the hyper-accumulation of starch. In addition, P_i_ uptake is optimized by increased expression of root P_i_ transporters and increased exudation of P_i_ solubilizing compounds, such as phosphatases, to release P_i_ organically bound to the rhizosphere. These adjustments are defined as systemic P_i_ deficiency responses [8]. Several factors mediating systemic P_i_ deficiency responses have been identified in recent years. The MYB transcription factors PHR1 (phosphate starvation response 1) and PHL (phosphate starvation response like) together with SPX proteins (named after SYG1 [supressor of yeast gpa1], PHO81 [CDK inhibitor in the yeast PHO pathway] and XPR1 [xenotropic and polytropic retrovirus receptor]) regulate the majority of P_i_ deficiency responses [2,9,10,11]. Under conditions of P_i_ sufficiency, the formation of complexes between SPX proteins and PHR1 or PHL proteins is promoted by P_i_ or phosphorylated compounds such as inositolphosphates, and prevents PHR1 or PHL from binding to the promoters of P_i_ deficiency response genes, thereby suppressing the induction of P_i_ starvation responses under conditions of P_i_ sufficiency [12,13,14]. Under P_i_ limiting conditions, decreasing levels of P_i_ or phosphorylated compounds lead to reduced formation of SPX-PHR1 and SPX-PHL complexes and the degradation of SPX proteins, enabling PHR1 and PHL binding to their target promoters and initiation of P_i_ deficiency responses. The basic helix-loop-helix type transcription factor bHLH32 was identified as a negative regulator of a subset of systemic P_i_ deficiency responses [15]. This factor was also shown to negatively regulate root hair formation in response to P_i_ starvation, indicating a role of bHLH32 in local P_i_ deficiency-responses: profound P_i_ deficiency-induced alterations in root system architecture resulting in increased formation of lateral roots and root hairs and in inhibition of primary root growth [16,17].

In dicotyledonous plants, Fe^3+^, the major form of iron in the soil, is first reduced and then taken up by the plant as Fe^2+^. Major phenotypic signs of Fe deficiency are yellowing of leaves provoked by chlorophyll loss and increased number of root hairs [18,19,20]. On the cellular level, there is a massive reprogramming of transcription, mainly of genes involved in iron uptake, distribution, and storage. Additionally, metabolic pathways leading to the production and exudation of Fe solubilizing compounds, such as malate, citrate, and coumarins, are induced [21,22,23,24,25,26,27,28]. These responses are activated by a complex multilayered regulatory network involving several bHLH type transcription factors. FIT (fer-like iron deficiency-induced transcription factor, bHLH29; [29,30] interacts with several subgroup Ib bHLH proteins [31], which, amongst others, are under the control of bHLH104 [20,32]. BHLH104 is also involved in the activation of PYE (popeye, bHLH47; [20,33]), which regulates several FIT-independent Fe deficiency responses. Furthermore, the protein level of bHLH104 is negatively regulated by BRUTUS (BTS), an E3 ligase containing a Fe binding hemerythrin domain, which relays the Fe status to the FIT-independent signal transduction network [20,34].

The regulatory networks controlling P_i_ or Fe deficiency responses have been comprehensively studied by comparing transcript profiles and phenotypes of WT plants and mutants with impaired expression of P_i_ and Fe deficiency response regulators. In contrast, there are only a few reports analyzing the contribution of P_i_ or Fe deficiency response regulators to P_i_ or Fe deficiency-induced metabolic changes. The comprehensive metabolite profiling study by Pant et al. [35] showed that P_i_ deficiency-induced accumulation of most metabolites was reduced in *phr1* mutants. However, the contribution of other P_i_ deficiency response regulators, such as PHL, SPX, or bHLH32 to metabolic adjustments has not been reported so far. In contrast to the broad metabolite profiling studies performed in the context of P_i_ deficiency [35,36], reports on metabolic changes in response to Fe deficiency are mainly restricted to the analysis of a few selected metabolites, such as malate, citrate, and coumarins [26]. Fe deficiency-induced accumulation of these compounds was shown to be mainly regulated by FIT, but the contribution of other Fe deficiency response regulators in this process has hardly been investigated. Only recently, Chutia et al. [37] analyzed the role of bHLH104, BRUTUS, and PYE in the formation of coumarin profiles in response to nutrient deficiencies.

Phosporus and iron form strong complexes, thereby limiting each others bioavailability. Especially under acid soil conditions, plant Fe content is increased with decreased P_i_ availability [38], suggesting that plant responses to P_i_ and Fe availability need to be coordinated. Considering the local P_i_ deficiency response, modulation of P_i_ deficiency-induced changes in root system architecture by Fe availability is well documented [16,17,39]. As such, P_i_ deficiency-induced inhibition of primary root growth and increased formation of lateral roots are strongly alleviated in the absence of Fe. Furthermore, the expression of many genes is regulated in an opposite manner by Fe and P_i_ deficiency and the promoters of many genes involved in Fe homeostasis contain the phosphate responsive P1BS (PHR1 binding sequence) domain [40]. On the metabolite level, Fe and P_i_ deficiency have opposite effects on coumarin accumulation in roots and their exudation into the rhizosphere [28,41], and Fe deficiency-induced accumulation of coumarins is suppressed by concomitant P_i_ deficiency [37]. However, effects of Fe availability on systemic P_i_ deficiency responses, such as shoot growth or root to shoot ratios or P_i_ deficiency-induced metabolic changes, which mainly affects metabolites in primary metabolism, have not been investigated so far.

In this study, we investigate shoot and root growth as well as metabolic changes in roots and shoots of *Arabidopsis thaliana* plants subjected to individual and combined P_i_ or Fe deficiencies. Since we were interested in the impact of Fe deficiency on systemic P_i_ deficiency responses, we focused our analysis on amino acids and organic acids levels. Phenotypic and metabolic responses of mutants were also investigated in order to assess the role of several P_i_ and Fe deficiency response regulators in the interdependent control of growth responses and of metabolite profiles by P_i_ and Fe availability.

## 2. Results

### 2.1. Phenotypic Changes of Col-0 Seedlings after P_i_, Fe, and Combined P_i_ and Fe Deficiency

Six days after transfer from nutrient sufficient conditions to P_i_ limitation, Col-0 seedlings exhibited the typical systemic P_i_ deficiency response exemplified by decreased root, shoot, and seedling fresh weights as well as increased root to shoot ratios, whereas transfer to Fe deficient conditions did not result in any changes in these growth parameters (Figure 1, Appendix A). Exposure of seedlings to P_i_ deficiency in the absence of Fe resulted in decreased shoot and seedling fresh weights and increased root to shoot ratios compared to seedlings grown under nutrient sufficient conditions. However, fresh weights and root to shoot ratios were significantly (*p* < 0.01) higher compared to seedlings transferred to P_i_ deficient conditions in the presence of Fe, indicating a positive effect of low Fe availability on seedling growth under conditions of suboptimal P_i_ availability.

### 2.2. Metabolic Changes in Col-0 Seedlings after P_i_, Fe, and Combined P_i_ and Fe Deficiency

In order to reveal whether the attenuation of the systemic P_i_ deficiency growth response under Fe deficient conditions also affects systemic P_i_ deficiency-induced metabolic changes, the concentrations of amino acids and organic acids in roots and shoots after treatment of Col-0 seedlings exposed to P_i_, Fe, and combined P_i_ and Fe deficiency (−Pi+Fe, +P_i_−Fe, −Pi−Fe, respectively) were analyzed. Principle component (PCA) analysis revealed a clear separation of samples exposed to P_i_ deficiency from samples transferred to nutrient sufficient conditions (Figure 2, Appendix A).

P_i_ deficiency treatment (−Pi+Fe) resulted in increased levels of almost all analytes (Figure 3, Appendix A). On average, these changes were more pronounced in shoots compared to roots, and were most prominent for amino acids containing one or more nitrogen atoms in their side chains, such as arginine, which accumulated 19- and 32-fold in roots and shoots, respectively. In contrast to P_i_ deficiency (−Pi+Fe), Fe deficiency (+P_i_−Fe) only marginally influenced metabolite levels in roots and shoots compared to untreated plants (+P_i_+Fe), as can be seen by an almost complete overlap in the PCA score plots (Figure 2). Fe deficiency-induced changes were very moderate with respect to the number of metabolites which exhibited significant (*p* ≤ 0.05; Student’s *t*-test, two tailed, equal variances) changes (about 50% of all analytes) as well as with respect to the extent of changes (Figure 3, Appendix A). The majority of analytes showing significant changes upon Fe deficiency exhibited slightly decreased levels by about 20%. Asparagine, leucine, and tyrosine in shoots as well as malate and citrate in roots represented the only metabolites displaying Fe deficiency-induced accumulation (Figure 3, Appendix A).

The PCA score plots further revealed separation of samples subjected to P_i_ deficiency in the presence of Fe (−Pi+Fe) from samples exposed to P_i_ deficiency in the absence of Fe (−Pi−Fe, Figure 2). This was more evident in roots compared to shoots. The plots also indicate that the metabolite profiles of samples treated with combined P_i_ and Fe deficiency (−Pi−Fe) were more similar to untreated (+P_i_+Fe) and −Fe treated (+P_i_−Fe) samples than to the metabolite profiles of samples exposed to P_i_ deficiency (−Pi+Fe). This was due to the reduction of P_i_ deficiency-induced amino acid levels by Fe deficiency, and was observed for all amino acids with the exception of alanine and glycine in roots and alanine, aspartate, asparagine, glycine, glutamate, and glutamine in shoots (Figure 3, Appendix A).

On average, Fe deficiency reduced P_i_ deficiency-induced amino acids levels by 30%. We also observed the trend that the negative effects of Fe deficiency were more pronounced for amino acids that exhibited strong P_i_ deficiency-induced accumulation, especially in shoots. Interestingly, of all analyzed metabolites, only malate and citrate further accumulated in response to combined Fe and P_i_ deficiency compared to either Fe or P_i_ deficiency alone. However, this could only be observed in roots.

### 2.3. Correlation between Metabolite Profiles and P_i_ Content

These results indicate overlapping and mostly opposing effects of Fe and P_i_ nutrition on seedling growth and primary metabolite content. The attenuation of P_i_ deficiency-induced responses by Fe limitation could be due to increased tissue P_i_ levels, which would suppress systemic P_i_ starvation responses. As seen in Figure 4 (Appendix A), Fe deficiency did not change endogenous P_i_ levels in plants in the presence of P_i_.

However, the reduced tissue P_i_ content observed after P_i_ deficiency was even further diminished by about 30% in roots and shoots under conditions of combined P_i_ and Fe deficiency. This suggests that the alleviation of P_i_ deficiency-induced growth inhibition by concomitant Fe limitation and the negative effect of decreased Fe availability on the accumulation of most metabolites by P_i_ deficiency occur independently from the endogenous P_i_ status of the plants. In order to verify this suggestion, mutants with impaired *pho1* expression, which exhibit an aberrant P_i_ distribution between roots and shoots [5,6,7] were analyzed (Figure 5, Appendix A). As expected, *pho1* plants exhibited lower and higher P_i_ content in shoots and roots, respectively, under conditions of P_i_ sufficiency. Accordingly, it was expected that metabolite levels in *pho1* plants are higher in shoots and lower in roots compared to Col-0 plants. This was observed for most metabolites in shoots. In roots, however, the content of only half of the analytes was affected, and only organic acids levels exhibited the expected decrease, whereas the levels of most amino acids showing significant changes (*p* ≤ 0.05; Student’s *t*-test, two tailed, equal variances) between *pho1* and Col-0 actually increased. Under combined P_i_ and Fe deficiency (−Pi−Fe), root P_i_ content was indistinguishable between Col-0 and *pho1* plants. However, *pho1* roots exhibited several alterations in metabolite levels. Most prominently, arginine and proline content was decreased by more than 50%, whereas citrulline levels more than doubled (Figure 5, Appendix A).

Under conditions of combined deficiency (−Pi−Fe), shoot P_i_ content of *pho1* plants was significantly higher compared to Col-0 plants. Accordingly, lower metabolite levels could have been expected. However, this only applied to some organic acids, such as fumarate, malate, and succinate, whereas no effect on amino acid levels was observed with the exception of glycine, which exhibited two-fold higher levels in *pho1* shoots compared to Col-0 shoots.

*Pho1* seedlings consistently exhibited lower tissue fresh weights compared to Col-0, even if increased tissue P_i_ concentrations were recorded. As such, root fresh weights were lower compared to Col-0 despite higher root P_i_ concentrations, and, although *pho1* seedlings exposed to combined P_i_ and Fe deficiency exhibited almost two-fold increased shoot P_i_ levels compared to Col-0, they exhibited reduced shoot fresh weights (Figure 6, Appendix A).

These results show that P_i_ deficiency results in comprehensive changes of plant growth and in metabolite profiles, and that these changes are strongly modulated by Fe availability. However, these modulations seem to be largely independent of internal P_i_ levels, suggesting regulation by additional, P_i_-independent factors. In order to possibly identify these additional factors, we analyzed the contribution of several known P_i_ and Fe deficiency response regulators in the changes of seedling growth and in the generation of metabolites profiles induced by P_i_, Fe and combined P_i_ and Fe deficiency.

### 2.4. Effect of P_i_ Deficiency Response Regulators on Metabolite Profiles

Mutants impaired in the expression of P_i_ deficiency response regulators profoundly affected metabolite profiles compared to Col-0 plants (Appendix A). Generally, the effects were more pronounced in shoots. In spite of indistinguishable growth compared to Col-0 (Figure 6), the mutants already showed strongly aberrant metabolite profiles in roots and shoots when seedlings were grown under nutrient sufficient (+P_i_+Fe) conditions. Under these conditions, bHLH32 and SPX1 as well as PHL1 seem to be the main factors controlling amino acid and organic acid levels in shoots and roots, respectively. Under conditions of P_i_ deficiency, mutants with aberrant *PHR1* expression revealed the strongest effects on metabolite profiles compared to Col-0 plants, although this mutant, like the other P_i_ deficiency response mutants, exhibited similar growth parameters compared to Col-0 (Figure 6). Although *phr1* exhibited mostly lower metabolite levels compared to WT under P_i_ deficiency conditions, it was still responsive to P_i_ deficiency, as seen by the accumulation of several metabolites, which was indistinguishable from WT (Figure 7). P_i_ deficiency-induced levels of some metabolites, such as asparagine and glutamine, were even higher in shoots of *phr1* compared to WT (Appendix A). *Phl1* plants, which are deficient in the expression of the PHR1 homologue PHL1, exhibited weaker effects on P_i_ deficiency-induced metabolite levels. Compared to the changes observed in *phr1* plants, a much lower number of metabolites was affected by the mutation in *PHL1* and the levels of these metabolites were less modified.

*Spx1* plants, which are deficient in the expression of the negative P_i_ deficiency response regulator SPX1, exhibited increased content of several metabolites upon P_i_ starvation compared to WT in roots. It is worth noting that most of these changes affected metabolites, which are not under the control of PHR1 or PHL1. Whereas *spx1* knock out plants exhibited higher metabolite concentrations after P_i_ deficiency predominantly in roots, *bhlh32* mutants showed higher metabolite concentrations after P_i_ deficiency predominantly in shoots. Interestingly, the absence of these two P_i_ deficiency response regulators more strongly affected metabolite profiles in response to Fe compared to P_i_ deficiency specifically in shoots (Appendix A). In case of *bhlh32*, the comparably strongly elevated metabolite concentrations were associated with increased root, shoot, and seedling fresh weights compared to Col-0 (Figure 6).

Since metabolite content differs between mutant and Col-0 plants grown under nutrient sufficient conditions, the comparison of only metabolite content between mutants and Col-0 plants exposed to nutrient limitations is not the appropriate means to elucidate the effect of P_i_ and Fe deficiency response regulators on the metabolic response upon nutrient deficiencies. Therefore, we determined differences in the responsiveness to the treatments between mutant and WT plants (Figure 7, Appendix A). Compared to WT, *phr1* plants exhibited most changes with respect to the number and the extent of P_i_ deficiency-induced metabolite accumulation. This was most evident in shoots. Although most analytes displayed reduced P_i_ deficiency-induced accumulation, some metabolites exhibited stronger accumulation in *phr1* plants compared to WT, e.g., ornithine in roots, and cysteine, aspartate, asparagine, glutamine, and threonine in shoots. Absence of PHL1 affected P_i_ deficiency-induced accumulation of only a few compounds, and most of these were also affected in *phr1* plants. However, some metabolites, such as glycine, serine, and valine, accumulated significantly less (two-way ANOVA *p* ≤ 0.05) compared to WT plants in *phl1* but not in *phr1* plants. Mutants impaired in the expression of the negative P_i_ deficiency response regulator SPX1 exhibited increased P_i_ deficiency-induced accumulation of metabolites in roots (Figure 7a, Appendix A). Remarkably, increased accumulation was observed for metabolites, such as isoleucine, leucine and asparagine, which were not affected in *phr1* or *phl1* mutants, although SPX1 exerts its negative effects by binding to PHR1 or PHL1. In shoots, mutation of *SPX1* led to decreased accumulation of some metabolites, such as glutamine and citrulline. Missing expression of bHLH32 only showed marginal effects on metabolite accumulation by P_i_ deficiency. However, *bhlh32* mutants exhibited several changes concerning the metabolic response to Fe deficiency, especially in shoots (Figure 7b, Appendix A). Compared to Col-0 plants, where Fe deficiency induced moderate depletion of metabolites, *bhlh32* mutants exhibited considerable accumulation of several metabolites. Interestingly, this was found mainly for metabolites, which were unaffected by Fe deficiency in Col-0 plants, such as isoleucine or arginine. In roots, *bhlh32* mutants responded to Fe deficiency by considerable decreases in glutamine and threonine concentrations (Figure 7b, Appendix A). In *phr1*, *phl1*, and *spx1* mutants, altered responsiveness to Fe deficiency was only sporadically observed for a few metabolites. Most evident was the considerable reduction in glycine and histidine levels by Fe deficiency in roots of *phl1* plants. The reduction in P_i_ deficiency-induced accumulation of metabolites by concomitant exposure of plants to Fe deficiency, which was observed in WT plants, was predominantly affected in mutants impaired in the expression of the P_i_ deficiency response regulators *phr1* and *spx1,* whereas the responses of *phl1* and *bhlh32* plants were almost indistinguishable from WT plants (Figure 7c, Appendix A). The negative effect of Fe deficiency on P_i_ deficiency-induced metabolite levels was mostly less pronounced or even absent in *phr1* and slightly enhanced in *spx1* plants, respectively. However, we also observed an enhancement of the negative effects of Fe deficiency in *phr1* plants for some metabolites, such as for asparagine in roots and for asparagine and glutamine in shoots. The weaker effect of Fe deficiency on P_i_ deficiency-induced metabolite levels in roots and shoots of *phr1* was accompanied by reduced fresh weights of both tissues compared to Col-0 under these conditions, whereas the change in the metabolic response to combined P_i_ and Fe deficiency in *spx1* was mainly restricted to roots, which exhibited increased fresh weight compared to WT roots (Figure 6).

### 2.5. Effect of Fe Deficiency Response Regulators on Metabolite Profiles

Mutants impaired in the expression of Fe deficiency regulators also strongly altered metabolites profiles (Appendix A). When grown in nutrient sufficient conditions, their impact on metabolite levels was even more pronounced compared to those observed in mutants deficient in the expression of P_i_ deficiency regulators (compare Appendix A, +P_i_+Fe). This was especially evident in shoots, where misexpression of *BRUTUS*, *FIT*,and *PYE* showed most deviations in metabolite profiles compared to WT plants of all mutants analyzed in this study. *Fit* and *pye* plants exhibited mainly decreased and increased, respectively, metabolite levels in roots as well as in shoots, whereas metabolite concentrations were predominantly increased in shoots, but decreased in roots of *brutus* mutants. *Bhlh104* mutants exhibited a lower number of changes compared to *fit*, *brutus*, and *pye* mutants, showing predominantly decreased and increased metabolite levels in roots and shoots, respectively, compared to WT plants. In spite of the strongly altered metabolite profiles in most of these mutants, only *pye* seedlings exhibited growth differences compared to Col-0, showing reduced root, shoot, and seedling fresh weights (Figure 6). Of all analyzed Fe deficiency response mutants, this mutant also exhibited the most striking metabolic changes compared to WT under conditions of P_i_ deficiency. It showed decreased and increased concentrations for amino and organic acids, respectively, in shoots, whereas root metabolite profiles were indistinguishable from Col-0 plants (Appendix A). Differences in growth under conditions of P_i_ deficiency could only be observed for seedlings of *fit*, which displayed increased shoot and seedling fresh weights compared to Col-0 (Figure 6). Under conditions of Fe deficiency, *pye* mutant plants exhibited increased metabolite levels compared to Col-0 plants in roots and shoots. This effect was much more pronounced in *fit* mutants. In this mutant, the concentrations of most amino acids were increased, whereas the concentrations of most organic acids, especially malate and citrate, were decreased. These changes were less prominent in *brutus* and *bhlh104* plants, and affected metabolite concentrations in roots as well as in shoots of *brutus* plants, but almost only in roots of *bhlh104* plants under conditions of Fe deficiency (Appendix A). In line with the comparably moderate metabolic changes, growth of *bhlh104* under conditions of Fe deficiency was indistinguishable from Col-0, whereas *brutus*, *fit*, and *pye* seedlings were impaired in shoot, root, and seedling growth compared to Col-0 (Figure 6).

In analogy to the mutants with impaired expression of genes coding for P_i_ deficiency response regulators, we determined differences in the responsiveness to the individual treatment between Fe deficiency response mutants and Col-0 plants in order to assess their contribution to nutrient deficiency-induced metabolic alterations (Figure 8, Appendix A).

The metabolic response induced by P_i_ starvation was different from WT plants mainly in *pye* mutants. In this mutant, P_i_ deficiency-induced accumulation was weaker or absent for several metabolites, in roots as well as in shoots. *Fit* and *brutus* plants showed predominantly lower accumulation of metabolites by P_i_ deficiency compared to WT plants, whereas *bhlh104* plants exhibited slightly stronger accumulation. However, these effects were observed for a much lower number of metabolites compared to *pye* plants (Figure 8a, Appendix A). Nutrient deficiency-induced changes in metabolite profiles were most different compared to WT plants under conditions of Fe deficiency. In this condition, *fit* plants exhibited the most differences, and responded to Fe deficiency by metabolite accumulation rather than by a reduction in metabolite content (Figure 8b, Appendix A), as was observed in Col-0 plants. Interestingly, the concentrations of malate and citrate in roots, which were the only compounds showing Fe deficiency-induced accumulation in Col-0 plants, declined or did not change, respectively, in this mutant. *Pye* mutants exhibited contrasting effects compared to *fit* plants, and displayed enhanced Fe deficiency-induced reduction in metabolite concentrations compared to WT plants. This was only observed in roots, whereas the response in *pye* shoots was almost indistinguishable from Col-0 plants (Figure 8b, Appendix A). Similarly, the Fe deficiency-induced metabolic response was altered in *bhlh104* plants only in roots, but not in shoots. In contrast, the metabolic response of *brutus* mutants to Fe deficiency was only different from Col-0 plants in shoots, but not in roots.

The negative effects of Fe deficiency on P_i_ deficiency-induced metabolite accumulation was hardly affected in *brutus* and *bhlh104* mutants (Figure 8c, Appendix A). In roots of brutus, reduction of P_i_ deficiency-induced tryptophan and tyrosine levels by concomitant Fe deficiency was less pronounced compared to WT plants, whereas Fe deficiency led to an enhanced reduction in P_i_ deficiency-induced asparagine and ornithine concentrations in roots and shoots, respectively, of *bhlh104* plants. The mutation in the Fe deficiency response regulator PYE also only marginally influenced the negative effect of Fe deficiency on P_i_ deficiency-induced metabolite profiles, showing increased or unaltered P_i_ deficiency-induced levels for aspartate and glutamate or citrulline and succinate, respectively, by Fe deficiency treatment in roots. In contrast, the reduction in P_i_ deficiency-induced metabolite levels by concomitant Fe deficiency was strongly compromised in plants impaired in the expression of FIT. However, the impact of this mutation was quite different between tissues. In roots, the negative effect of Fe deficiency on P_i_ deficiency-induced accumulation of metabolites was absent for several metabolites, whereas it was even enhanced for several metabolites in shoots (Figure 8c, Appendix A). Whereas the absence of a metabolic response to combined deficiency in roots of *fit* seedling had no consequences on root growth, the increased reduction of P_i_ deficiency-induced metabolite accumulation by Fe deficiency in shoots was accompanied by a strongly increased shoot fresh weight compared to Col-0, which resulted in a decreased root to shoot ratio (Figure 6).

## 3. Discussion

Phosphate deficiency induces a profound reprogramming of cellular metabolism enabling the plant to cope with the low availability of P_i_. This includes mechanisms to maintain P_i_ homeostasis by the reduction of P_i_ consuming metabolic activities and by prioritization of metabolic pathways which releases P_i_. The accumulation of amino acids represents one prominent metabolic response to P_i_ deficiency. Our results showing the accumulation of almost all amino acids during exposure to P_i_ limiting conditions corroborate previous reports [35,36]. There are only minor, mainly quantitative, differences between the studies. The stronger accumulation of amino acids reported in the study by Pant et al. [35] is probably due to the initial growth of plants on low P_i_ containing medium before transfer to P_i_ depleted conditions, whereas the weaker accumulation observed by Morcuende et al. [36] might be due to a shorter exposure of plants to P_i_ depleted conditions. Nevertheless, amino acid accumulation can be considered as a very robust and consistent marker of P_i_ deficiency response. However, the physiological significance of amino acid accumulation under conditions of insufficient P_i_ supply is not entirely clear. Elevated amino acid concentrations as a result of inhibition of protein synthesis in combination with increased protein degradation have been suggested [35,36]. However, increased transcript levels of genes involved in amino acid biosynthesis also suggests that P_i_ deficiency-induced amino acid accumulation is due to increased amino acid biosynthesis [12,35,36]. Increased amino acid biosynthesis, especially of N-rich amino acids, such as histidine, tryptophan, and arginine, was discussed as a mechanism to assimilate and transport ammonia, which might accumulate to toxic levels by continuous nitrate reduction and by an increased rate of photorespiration under P_i_ limiting conditions [35]. According to our data, this does not seem to be decisive for plant growth under P_i_ limiting conditions, since *fit* mutants exhibited increased shoot and seedling fresh weight compared to Col-0, although the content as well as P_i_ deficiency-induced accumulation of N-rich amino acids were lower in *fit* compared to Col-0 (Figure 6, Figure 8 and Appendix A). Under conditions of combined P_i_ and Fe deficiency, increased levels of N- rich amino acids in *fit* roots might indicate better plant growth of this mutant because of improved assimilation of nitrate. However, the concentrations in shoots of these amino acids are lower compared to Col-0 despite improved shoot growth. It is generally difficult to elucidate a causal relationship between the concentrations of individual metabolites and plant growth. Our data only allow to infer a trend that metabolite concentrations and growth show an inverse correlation, such as in Col-0 seedlings grown under P_i_ limiting conditions as well as in *brutus*, *fit*, and *pye* mutants exposed to Fe deficiency, or in *fit* mutants subjected to P_i_ limiting conditions in the presence as well as in the absence of Fe (Figure 6, Figure 8 and Appendix A). However, this does not apply to all mutants, since *bhlh32* plants showed increased growth and metabolite concentrations under conditions of Fe deficiency, whereas *phr1* seedlings exhibited decreased growth and metabolite levels compared to Col-0 under conditions of combined deficiency (Figure 6, Figure 7 and Appendix A). Considering the multifaceted mechanisms determining plant growth, it is hardly possible to reason phenotypic plasticity of plants subjected to nutrient deficiencies by following the changes of a subset of metabolites, such as in this study. However, the strong and divergent changes in amino and organic acid levels of roots and shoots in response to various nutrient limitations in Col-0 and several mutants enabled us to provide a deeper understanding of the regulation of metabolic responses to nutrient limitations.

### 3.1. Regulation of P_i_ Deficiency-Induced Metabolite Profiles

Pant et al. [35] reported that amino acid accumulation was dependent on the MYB transcription factor PHR1, which is known to regulate more than 70% of P_i_ deficiency-induced transcriptional reprogramming [9]. Our study also indicates a major role of PHR1 in the accumulation of amino acids under conditions of P_i_ deficiency (Figure 7). However, according to our results, PHR1 seems to regulate the accumulation of only a subset of amino acids, especially in roots. In agreement with the study by Pant et al. [35], mutations in PHR1 did not completely abolish P_i_ deficiency-induced metabolite accumulation, indicating the cooperation by additional regulators, such as PHL1, which was shown to act in concert with PHR1 in the transcriptional reprogramming during P_i_ deficiency [9]. This cooperativity was also evident based on our metabolite profiles showing that mutations in PHL1 and PHR1 affected P_i_ deficiency-induced accumulation of an overlapping set of metabolites. However, reduced accumulation of several amino acids, such as glycine, serine, and valine, in *phl1* but not in *phr1* mutants also suggests distinct targets for both transcription factors. Furthermore, other factors besides PHR1 and PHL1 are probably involved in the regulation of P_i_ deficiency-induced metabolic adjustments, since accumulation of several metabolites was affected neither in *phr1* nor *phl1* mutants. Other PHR-like transcriptional regulators, of which 15 isoforms have been described in Arabidopsis [2,11], are likely candidates, and it will be interesting in the future to elucidate their contribution and specificity in metabolic reprogramming during P_i_ limitation.

The role of SPX proteins as negative regulators of P_i_ deficiency responses was also evident based on our metabolite profiles; however, mutants impaired in the expression of SPX1 showed increased accumulation of only a few amino acids exclusively in roots (Figure 7). Additionally, SPX1 seems to be involved in the regulation of P_i_ deficiency-induced accumulation of some metabolites, such as isoleucine, leucine, or asparagine, which are not subject to regulation by PHR1 or PHL1. Considering that SPX proteins exert negative regulation of P_i_ deficiency responses by binding to PHR1 and probably also PHL proteins [12], our results suggest complex and selective regulation of P_i_ deficiency-induced metabolic adjustments, which is accomplished by formation of specific interactions between SPX isoforms, of which four isoforms are described in Arabidopsis [10], and PHR1 or PHL isoforms. Additionally, our data showing tissue specificity of metabolic alterations in *spx1*, *phr1*, and *phl1* mutants indicate that this interaction selectivity might vary in roots and shoots (Figure 7a).

The bHLH transcription factor bHLH32 seems to play a minor role in the regulation of metabolic alterations under conditions of P_i_ limitation. BHLH32 was identified as a negative P_i_ deficiency response regulator based on increased root hair formation and increased induction of the phospho*enol*pyruvate carboxylase pathway under conditions of P_i_ limitation in *bhlh32* mutants [15]. Curiously, this factor seems to play a considerable role with respect to Fe deficiency, especially in shoots, where *bhlh32* mutants exhibited very different metabolic Fe deficiency responses compared to WT (Figure 7b). These data combined with the fact that root hair formation and induction of phospho*enol*pyruvate carboxylase pathway genes are prominent Fe deficiency responses [19,21,22,33,40,43] instead suggest a role for bHLH32 in the regulation of Fe deficiency responses.

On the other hand, absence of PYE resulted in the suppression of P_i_ deficiency-induced accumulation of several metabolites (Figure 8a). This bHLH type transcription factor is involved in the regulation of several responses to Fe deficiency [33]. Considering metabolism, genes involved in the biosynthesis of the metal chelator nicotianamine were identified as a major target of PYE. Furthermore, microarray analysis revealed deregulation of several genes involved in amino acid metabolism in *pye* mutants under nutrient sufficient conditions, such as asparagine synthase, branched chain aminotransferase, arogenate dehydratase, or ATP-phosphoribosyl transferase [33]. This is reflected by considerable differences in metabolite concentrations between *pye* mutant and WT plants when grown in nutrient sufficient conditions (Appendix A). Therefore, the peculiar metabolic P_i_ deficiency response in this mutant might be a consequence of generally disturbed metabolic activities. However, *fit* as well as *brutus* mutants hardly showed changes in the metabolic responses to P_i_ deficiency, although they also revealed strong alterations in metabolite profiles compared to WT plants already under control conditions. As such, a specific role of PYE in the regulation of P_i_ deficiency-induced metabolic adjustments cannot be excluded. How this Fe deficiency regulator integrates into the existing regulatory network controlling P_i_ deficiency responses remains an interesting question for future experiment.

### 3.2. Regulation of Fe Deficiency-Induced Metabolite Profiles

The metabolic response to Fe deficiency was most different in *fit* mutants compared to WT, which emphasizes the importance of this major regulator of Fe deficiency responses [29] also in metabolic adjustments (Figure 8). Most importantly, our data show that FIT regulates the accumulation of malate and citrate in roots. Following exudation by roots of Fe starved plants, both organic acids were shown to chelate Fe bound to the rhizosphere leading to an increased bioavailability of Fe [44]. Mutants impaired in the expression of bHLH104 showed similar trends in the metabolic alterations compared to *fit* mutants, although the effects are milder with respect to the number and the extent of changes. This corroborates the current model, in which initiation of FIT dependent Fe deficiency responses requires the interactions of bHLH104 with several other bHLH proteins, but in which FIT dependent Fe deficiency responses can still be initiated in the absence of bHLH104, although to a lower extent [20,32].

The stronger metabolic response to Fe deficiency in *pye* mutants compared to Col-0 agrees with the proposed role of PYE as a negative regulator of a subset of Fe deficiency responses [33]. Interestingly, the Fe binding E3 ligase BTS plays only a minor role in the generation of Fe deficiency-induced metabolite profiles, although it was shown to coordinate FIT and PYE mediated Fe deficiency signaling by targeting bHLH104 and its interaction partners for degradation [33,45]. This suggests the existence of other Fe sensing proteins, which are involved in Fe dependent metabolic adjustments. Likely candidates are BRUTUS-LIKE E3 ligases, which are known to target FIT for degradation [46], since *fit* mutants exhibited most effects on metabolite profiles of all analyzed Fe deficiency response regulators. Alternatively, tissue Fe status might not represent the major determinant for the generation of metabolite profiles under conditions of low Fe supply because the effect of Fe deficiency on P_i_ deficiency-induced metabolite accumulation was PYE, bHLH104, BTS, and FIT-independent for the majority of the analyzed compounds (Figure 8c). A possible alternative mechanism leading to the negative effect of Fe deficiency on P_i_ deficiency-induced metabolite accumulation by will be discussed in more detail in the following paragraph.

### 3.3. Modulation of P_i_ Deficiency-Induced Metabolite Profiles by Fe Deficiency

Our data showing a strong negative effect of Fe deficiency on P_i_ deficiency-induced metabolite accumulation raise the question about the mechanism underlying this effect, especially since Fe deficiency alone only mildly affects the metabolite profiles (Figure 3). As mentioned above, the fact that mutations in bHLH104, BRUTUS, or PYE hardly modified this response might imply that modulation of metabolite profiles by Fe deficiency is not a direct effect of lower tissue Fe content. It is conceivable that low Fe leads to reduced formation of Fe−Pi complexes, thereby elevating tissue P_i_ concentration in tissues, which would alleviate the metabolic P_i_ deficiency response. This scenario is not supported by our data showing that P_i_ levels are rather decreased under conditions of combined deficiency (Figure 4) as well as by the experiments using *pho1* mutants, in which no correlation between aberrant free P_i_ distribution and metabolic responses was detectable (Figure 5). Recent studies point to a more important role for the inositol pyrophosphates IP7 and IP8 rather than for free P_i_ as mediators of PHR or PHL interactions with SPX proteins in the suppression of systemic P_i_ deficiency responses [13,14]. Lowering endogenous Fe content by low external Fe supply could increase the level of free inositol 6-phosphate (IP6, phytate), a very potent Fe chelator [47], leading to increased substrate availability for the biosynthesis of IP7 and IP8, which exhibit the strongest affinity to SPX proteins [13]. Increased IP7 and IP8 formation in the absence of Fe could lead to enhanced formation of complexes between PHR or PHL and SPX proteins, preventing PHR or PHL from activating P_i_ deficiency responses. However, the negative effect of Fe deficiency on P_i_ deficiency-induced metabolite accumulation was only partially altered in *phr1*, *phl1*, and *spx1* mutants (Figure 7c). Furthermore, transcriptome data do not reveal Fe deficiency-induced changes in transcript levels of genes involved in the biosynthesis of IP7 and IP8 [21,22,33,40,43], such as ITPK1, ITPK2, VIH1, and VIH2 [14,48]. Therefore, the mechanism, by which Fe suppresses P_i_-deficiency responses, remains obscure, since the present data neither support signaling pathways which are initiated by reduced Fe content or by increased levels of P_i_ or inositol pyrophosphates. More comprehensive analyses of mutants, including more *phl* or *spx* mutants as well as more mutants deficient in the expression of Fe deficiency response regulators combined with studies of the transcriptome, the proteome, and the metabolome are clearly needed to decipher the molecular components coordinating P_i_ and Fe deficiency response signaling pathways.

### 3.4. Physiological Relevance of P_i_ and Fe Deficiency-Induced Metabolic Adjustments

An important question concerns the physiological relevance of decreased primary metabolite levels after Fe deficiency and the negative impact of Fe deficiency on P_i_ deficiency-induced metabolite accumulation. For combined deficiency, a general breakdown of metabolism under conditions of severe malnutrition of plants with respect to Fe and P_i_ could be discussed. However, plants were harvested before shoots exhibited visible signs of Fe deficiency, such as leaf chlorosis, P_i_ deficiency-induced metabolite accumulation was still present in the absence of Fe, and tissue fresh weights were higher compared to P_i_ deficiency in the presence of Fe. Furthermore, omission of Fe leads to an alleviation of P_i_ deficiency-induced primary root growth arrest [16,17], and Fe deficiency-induced coumarin accumulation was still detectable in P_i_ starved plants [37]. These findings indicate active metabolism also in case plants suffer from P_i_ and Fe starvation.

Possibly, Fe deficiency could lead to the allocation of primary metabolites to the synthesis of compounds which enable plants to maintain Fe homeostasis and to increase Fe acquisition. As such, decreased levels of aromatic amino acids as well as of Met, Arg, citrulline, and ornithine might be due to the Fe deficiency-induced accumulation of coumarins [22,24,25,26,27,28,37,49], nicotianamine [50], or putrescine [51]. However, we could only detect Fe deficiency dependent depletion of all these amino acids under conditions of low P_i_ supply. Possibly, the levels of these amino acids are high enough only at low P_i_ conditions so that they can be metabolized without biosynthetic replenishment, whereas, under conditions of high P_i_, the consumption of these amino acids must be compensated by increased biosynthesis in order to maintain plant survival. This assumption might be supported by the fact that of all genes involved in amino acids metabolism, which are regulated by Fe deficiency, those implicated in the metabolism of aromatic amino acids, as well as of Met, Arg, citrulline, and ornithine, are overrepresented [21,22,33,40,43]. However, considering the strong allosteric regulation of primary metabolism, it is questionable whether the analysis of transcript levels is sufficient to estimate Fe deficiency-induced reprogramming of primary metabolism. In that respect, a more comprehensive analysis of primary metabolites as well as metabolic flux analysis is required to substantiate the assumption of increased utilization and consumption of amino acids as building blocks for the generation of compounds, which are known to alleviate Fe deficiency.

A further physiological role for the observed Fe deficiency-induced metabolite profiles could be inferred from the study by Zhu et al. [52]. They showed the pivotal role of ammonium accumulation in the regulation of Fe deficiency responses. It is conceivable, that Fe deficiency-induced depletion of amino acids, either by decreased biosynthesis or increased degradation, could result in the generation of ammonium, which was shown to upregulate genes involved in Fe uptake and translocation resulting in increased soluble Fe content and decreased sensitivity to Fe deficiency [52]. Thus, the contrasting effects of Fe and P_i_ deficiency on amino acid levels observed in this study could be interpreted as a strategy to balance the levels of ammonium, which is required to decrease sensitivity to Fe deficiency but which needs to be detoxified under conditions of P_i_ deficiency [35]. Clearly, more experimental evidence about the role of ammonium during P_i_ and Fe deficiency and about the biogenesis of ammonium under these conditions is needed to substantiate this assumption.

Interestingly, P_i_ and Fe deficiency act synergistically in the accumulation of the organic acids malate and citrate in roots. Both compounds are known to facilitate P_i_ and Fe nutrition, either by their potential to solubilize both nutrients from insoluble complexes in the rhizosphere or by mobilization of Fe. Possibly, increased plant growth under conditions of combined deficiencies compared to P_i_ deficiency might be a result of increased P_i_ acquisition mediated by the additional accumulation of malate caused by Fe deficiency. Vice versa, increased Fe acquisition and mobilization might result from the additional accumulation of citrate induced by P_i_ deficiency. Indeed, Fe content was shown to increase under conditions of P_i_ deficiency [38,53]. Combined P_i_ and Fe deficiency treatment did not elevate P_i_ concentrations compared to single P_i_ deficiency. However, it is possible that a larger proportion of P_i_ is organically bound in order to sustain plant growth, resulting in higher total P content. In this context, it would be interesting in the future to determine the level of total P and of phosphorylated compounds as well as to analyze P_i_ uptake.

In summary, our results clearly reveal that the strong and mainly contrasting effects of P_i_ and Fe deficiency are not only restricted to phenotypic responses, such as P_i_ deficiency-induced root growth inhibition [16,17], or to Fe deficiency-induced accumulation of secondary compounds [37]. Instead, P_i_ and Fe availability also lead to a mostly opposing reprogramming of primary metabolism, which is coordinated by the action of several P_i_ and Fe deficiency response regulators. It would be interesting in the future to elucidate the mechanisms underlying this complex regulation on the molecular level with respect to the interaction of P_i_ and Fe deficiency response regulators as well as with respect to the identification of the molecules or of the ratios of molecules, which are sensed and which initiate the signaling cascades.

## 4. Materials and Methods

### 4.1. Plant Lines and Growth Conditions

T-DNA insertion lines were obtained from the Nottingham Arabidopsis Stock Center (NASC, University of Nottingham, England, UK), and were all in the *Arabidopsis thaliana* accession Col-0 background, which was used as WT throughout the study. Homozygosity of the plants was confirmed as described [37].

Growth conditions were exactly as described [37] with some smaller modifications. Briefly, seeds were surface sterilized with chlorine gas and placed on sterile agar plates (1% (*w*/*v*) Phyto Agar, Duchefa, Haarlem, The Netherlands, purified according to [16]) containing 5 mM KNO_3_, 0.5 mM KH_2_PO_4_, 2 mM MgSO_4_, 2 mM Ca(NO_3_)_2_, 50 µM Fe-EDTA, 70 µM H_3_BO_3_, 14 µM MnCl_2_, 0.5 µM CuSO_4_, 1 µM ZnSO_4_, 0.2 µM Na_2_MoO_4_, 10 µM CoCl_2_, and 5 g L^−1^ of sucrose buffered with 2.5 mM Mes-KOH to pH 5.6. For −Fe medium, Fe-EDTA was omitted, and, for –P_i_ medium, the concentration of KH_2_PO_4_ was reduced to 5 µM. Including the P_i_ and Fe amount originating from the purified agar, the final concentrations of P_i_ and Fe amounted to 6 µM and 1 µM in −Pi and −Fe medium, respectively. Seed germination was synchronized by incubation of the plates for 2 days in the dark at 4 °C. Afterwards, agar plates were incubated in a vertical position in a growth chamber at 22 °C under illumination for 16 h daily (170 µmol s^−1^ m^−2^; Osram LumiluxDeLuxe Cool daylight L58W/965, Osram, Augsburg, Germany). After 5 days of growth in nutrient sufficient conditions (+P_i_,+Fe), seedlings were transferred to fresh agar plates containing the respective conditions (+P_i_,+Fe; −Pi,+Fe; +P_i_,−Fe; −Pi,−Fe). After an additional 6 days of growth, roots were separated from the shoots, their fresh weight recorded, and frozen in liquid nitrogen until further processing. Average root, shoot, and seedling fresh weights as well as root to shoot ratios of Col-0 seedling exposed to all four conditions are shown in Appendix A. The root, shoot, and seedling fresh weights as well as root to shoot ratios of all genotypes and treatments are listed in Appendix A. Photographs of seedlings at the time of harvest are shown in [37]. For metabolite analysis, one biological replicate consisted of roots or shoots from two plants.

### 4.2. Metabolite Analysis

Tissues (1–5 mg of fresh weight) were ground using 5 mm steel beads in a bead mill at 25 Hz for 50 s, and the resulting powder was extracted by vigorous shaking for 20 min with 100 µL of 70% (*v/v*) methanol containing as internal standards 2 nmol norvaline (amino acid quantification), 5 nmol [2,2,3,3-^2^H] succinic acid (P_i_, fumarate, succinate, and 2-oxoglutarate quantification), 5 nmol [2,2,3,-^2^H] malic acid (malate quantification), and [2,2,4,4-^2^H] citric acid (citrate and aconitate quantification). Clear extracts were obtained by two rounds of centrifugations (5 min, 12,000× *g*). Targeted amino acid profiling was performed by LC-ESI-MS/MS consisting of a 1290 LC system (Agilent, Waldbronn, Germany) coupled to an API 3200 triple quadrupole mass spectrometer (AB Sciex, Darmstadt, Germany) as described [54] using 25 µL of the extracts. For determination of organic acid and P_i_ content, 10 µL of the extracts were dried, methoxylated for 1.5 h at room temperature with 20 µL of 20 mg mL^−1^ of methoxyamine in pyridine (Sigma-Aldrich, St. Louis, MO, USA), and silylated for 30 min at 37 °C with 35 µL of Silyl 991 (Macherey-Nagel, Düren, Germany). Gas chromatography (GC)-MS/MS analysis was performed as described [37]. The Agilent 7890 GC system was equipped with an OPTIMA 5 column (10 m × 0.25 mm, 0.25 µm; Macherey-Nagel, Düren, Germany) and coupled to an Agilent 7000B triple quadrupole mass spectrometer (Agilent, Waldbronn, Germany) operated in the positive chemical ionization mode (reagent gas: methane, gas flow: 20%, ion source temperature; 230 °C). One microliter was injected [pulsed (25psi) splitless injection] at 220 °C. After 1 min at 60 °C, the temperature was initially increased at 35 °C min^−1^ to 200 °C and finally at 50 °C min^−1^ to 340 °C. The final temperature of 340 °C was held for 5 min. Helium was used as the carrier at 2.39 mL min^−1^. The transfer line was set to a temperature of 250 °C. Helium and N_2_ were used as quench and collision gases, respectively (2.25 and 1.5 mL min^−1^). Multiple reaction monitoring parameters for the detection of the metabolites are indicated in Appendix A. The IntelliQuant algorithm of the Analyst 1.6.2 software (AB Sciex, Darmstadt, Germany) or the Agile algorithm of the MassHunter Quantitative Analysis software (version B06.00, Agilent, Waldbronn, Germany) were used to integrate the peaks for amino acids or organic acids and P_i_, respectively. Metabolite concentrations were calculated using the respective internal standards and divided by the fresh weights. In order to account for variations in absolute metabolite concentrations between independent experiments, all values within individual experiments were normalized to the average values of the biological replicates of the Col0 +P_i_+Fe treatment in the respective experiment. Average absolute metabolite concentrations in roots and shoots of Col-0 plants grown under nutrient sufficient conditions (+P_i_+Fe) are listed in Appendix A. Average absolute and relative metabolite concentrations for all treatments and genotypes can be found in Appendix A.

### 4.3. Principle Component Analysis

Data was analyzed using R version 3.6.2, on a Linux ×86_64 server with Debian GNU/Linux 10. The data were mean centered, and PCA calculated with prcomp() in the stats package. The entire analysis behind Figure 2, including CSV data files, is available as R vignette and HTML as Appendix A and at 10.5281/zenodo.4337054.

## Figures and Tables

**Figure 1 ijms-22-07609-f001:**
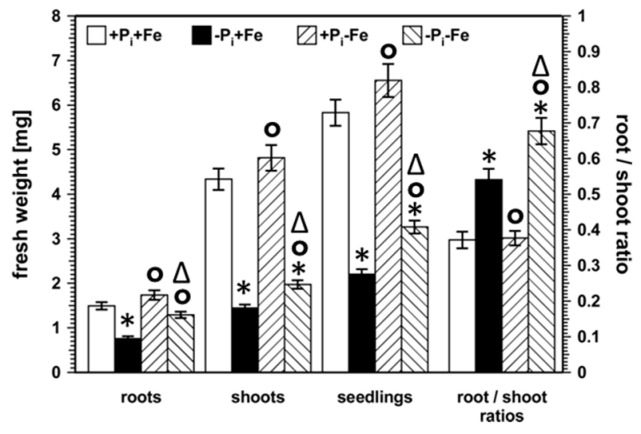
Root, shoot, and seedling fresh weight as well as root to shoot ratios of Col-0. After five days on nutrient sufficient conditions (+P_i_+Fe), seedlings were transferred to the indicated conditions and allowed to grow for an additional 6 days before harvest. +P_i_: 500 µM, −Pi: 5 µM, +Fe: 50 µM, −Fe: no Fe added. Error bars indicate SE (*n* = 40). Significance analyses between treatments were performed by Student’s *t*-test (two tailed, equal variances). * *p* < 0.01 compared to +P_i_+Fe; ° *p* < 0.01 compared to −P_i_+Fe; ^∆^ *p* < 0.01 compared to +P_i_−Fe. The respective numbers are shown in Appendix A.

**Figure 2 ijms-22-07609-f002:**
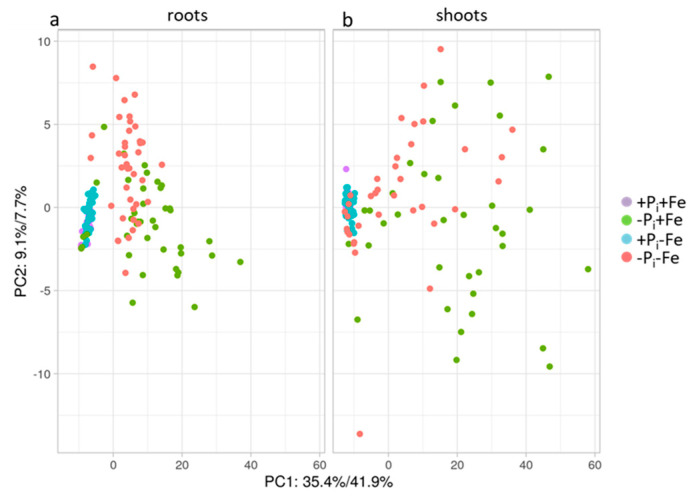
PCA score plots of the different treatments, (**a**) Col-0 roots; (**b**) Col-0 shoots. After an initial growth period of five days on nutrient sufficient conditions (+P_i_+Fe), seedlings were transferred to the indicated conditions and allowed to grow for an additional six days before harvest. +P_i_: 500 µM, −Pi: 5 µM, +Fe: 50 µM, −Fe: no Fe added.

**Figure 3 ijms-22-07609-f003:**
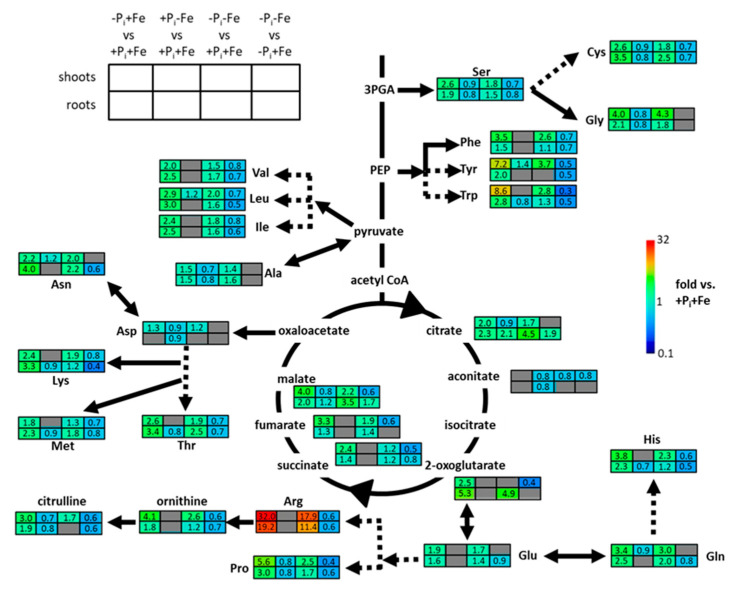
Metabolic changes in Col-0 plants presented in the form of a core metabolism overview, modified based on MAPMAN [42]. Upper and lower rows indicate the normalized metabolite concentrations (normalized to +P_i_+Fe) in shoots and roots, respectively, after treatment with P_i_ (−Pi+Fe), Fe (+P_i_−Fe) and combined P_i_ and Fe (−Pi−Fe) deficiencies (columns from left to right). Columns to the right represent the ratios between metabolite levels after combined P_i_ and Fe deficiency (−Pi−Fe) and P_i_ deficiency (−Pi+Fe). Cells colored in gray indicate no difference to plants grown under nutrient sufficient conditions (+P_i_+Fe) or between −Pi−Fe and −Pi+Fe treatments at *p* ≤ 0.05 (Student’s *t*-test, two tailed, equal variances, *n* ≥ 27). The numbers refer to the values shown in Appendix A.

**Figure 4 ijms-22-07609-f004:**
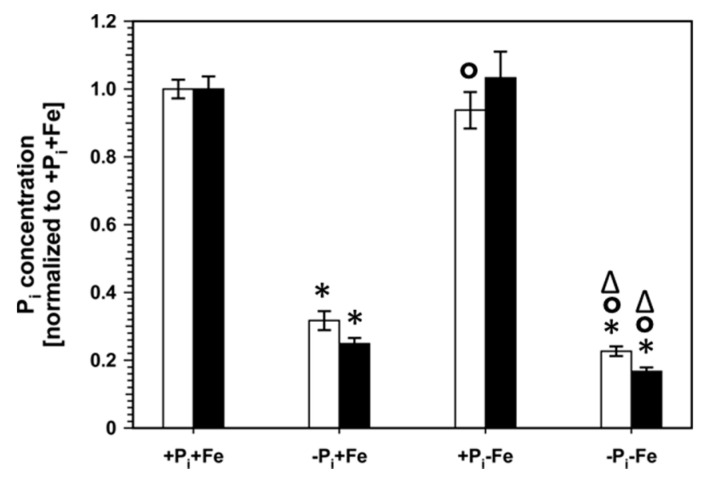
P_i_ concentrations in roots (open bars) and shoots (closed bars) of Col-0 plants. After an initial growth period of five days on nutrient sufficient conditions (+P_i_+Fe), seedlings were transferred to the indicated conditions and allowed to grow for an additional six days before harvest. +P_i_: 500 µM, −Pi: 5 µM, +Fe: 50 µM, −Fe: no Fe added. Data are normalized to Col-0 +P_i_+Fe, average absolute amounts: roots: 10.7 nmol mg^−1^ FW; shoots: 7.6 nmol mg^−1^ FW. Error bars indicate SE (*n* ≥ 27). Significance analyses between treatments were performed by Student’s *t*-test (two tailed, equal variances). * *p* ≤ 0.05 compared to +P_i_+Fe; ° *p* ≤ 0.05 compared to −Pi+Fe; ^∆^ *p* ≤ 0.05 compared to +P_i_−Fe. The respective numbers are shown in Appendix A.

**Figure 5 ijms-22-07609-f005:**
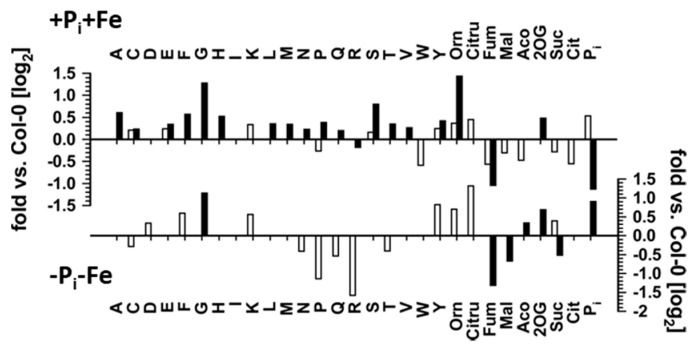
Metabolic changes in roots (open bars) and shoots (closed bars) of *pho1* mutants relative to Col-0 plants after transfer to nutrient sufficient (+P_i_+Fe) conditions (upper graph) or to conditions of combined P_i_ and Fe nutrient deficiencies (−Pi−Fe, lower graph). After an initial growth period of five days on nutrient sufficient conditions (+P_i_+Fe), seedlings were transferred to +P_i_+Fe or −Pi−Fe and allowed to grow for an additional six days before harvest. +P_i_: 500 µM, −Pi: 5 µM, +Fe: 50 µM, −Fe: no Fe added. Bars denote the log_2_ fold change in metabolite levels between *pho1* and Col-0. Only changes at *p* ≤ 0.05 (Student’s *t*-test two tailed, equal variances) are shown (*n* ≥ 8). The corresponding data are listed in Appendix A. Orn: ornithine; Citru: citrulline; Fum: fumarate; Mal: malate; Aco: aconitate; 2OG: 2-oxoglutarate; Suc: succinate; Cit: citrate.

**Figure 6 ijms-22-07609-f006:**
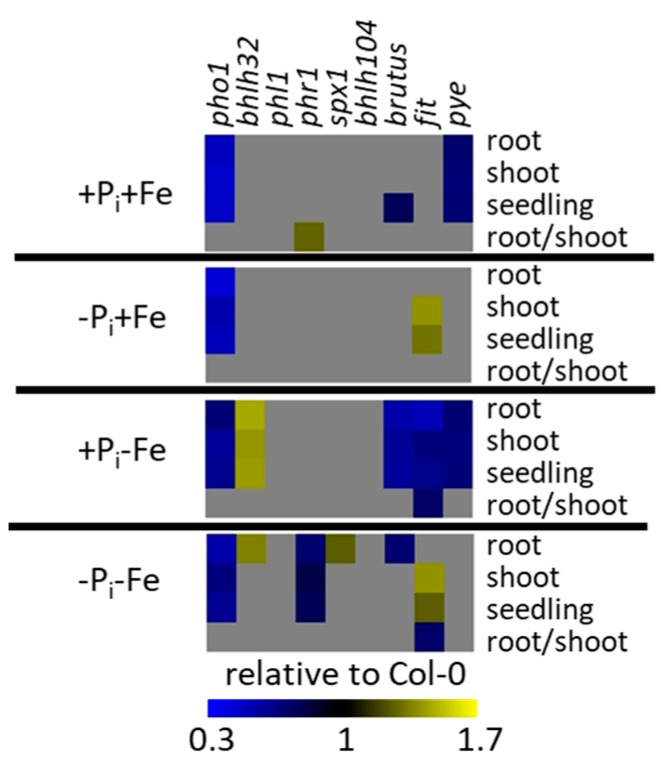
Root, shoot, and seedling fresh weights as well as root to shoot ratios in mutants with impaired expression of P_i_ and Fe deficiency response regulators relative to Col-0. Grey boxes indicate no significant differences (*p* ≤ 0.05, Student’s *t*-test, two tailed, equal variances, *n* ≥ 8) between mutants and Col-0. After an initial growth period of five days on nutrient sufficient conditions (+P_i_+Fe), seedlings were transferred to the indicated conditions and allowed to grow for an additional six days before harvest. +P_i_: 500 µM, −Pi: 5 µM, +Fe: 50 µM, −Fe: no Fe added. Heat maps were generated according to the data shown in Appendix A.

**Figure 7 ijms-22-07609-f007:**
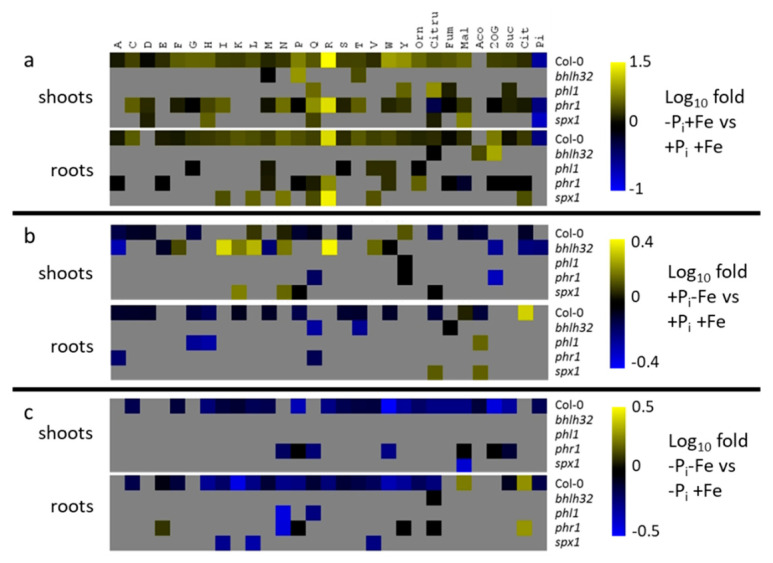
Metabolite changes in shoots (upper panels) and roots (lower panels) of Col-0 plants and mutants with impaired expression of P_i_ deficiency response regulators. (**a**) response to P_i_ deficiency in the presence of Fe (−Pi+Fe vs. +P_i_+Fe); (**b**) response to Fe deficiency in the presence of P_i_ (+P_i_−Fe vs. +P_i_+Fe); (**c**) response to combined P_i_ and Fe deficiency compared to P_i_ deficiency (−Pi−Fe vs. −Pi+Fe). After an initial growth period of five days on nutrient sufficient conditions (+P_i_+Fe), seedlings were transferred to the indicated conditions and allowed to grow for an additional six days before harvest. +P_i_: 500 µM, −Pi: 5 µM, +Fe: 50 µM, −Fe: no Fe added. The log_10_ fold changes are color coded. For Col-0, only changes at *p* ≤ 0.05 (Student’s *t*-test, two tailed, equal variances) are color coded. For the mutants, changes are color coded, if the metabolic response was different compared to the WT response at *p* ≤ 0.05 (two way ANOVA, *n* ≥ 8). Grey boxes indicate no change in metabolite concentrations in Col-0 by the respective treatment, or no difference in the metabolic response to the respective treatments between mutants and WT. Orn: ornithine; Citru: citrulline; Fum: fumarate; Mal: malate; Aco: aconitate; 2OG: 2-oxoglutarate; Suc: succinate; Cit: citrate. Heat maps have been generated according to the data shown in Appendix A.

**Figure 8 ijms-22-07609-f008:**
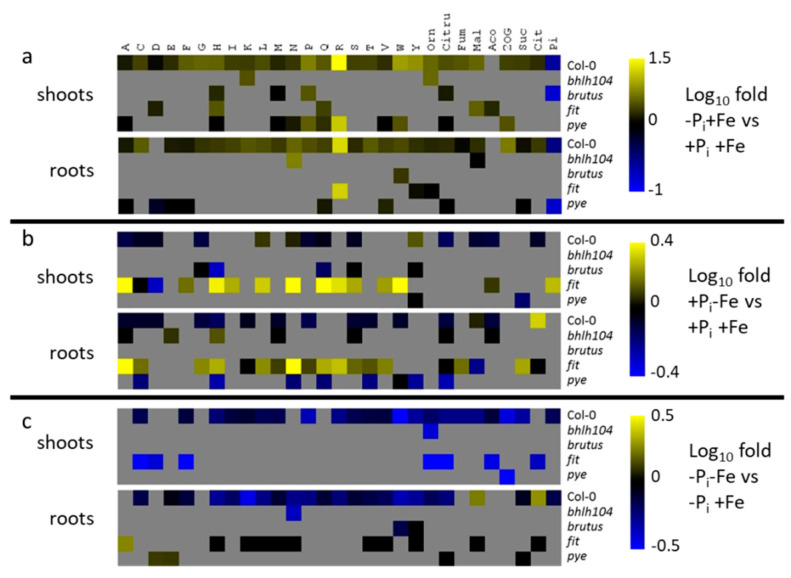
Metabolite changes in shoots (upper panels) and roots (lower panels) of Col-0 plants and mutants with impaired expression of Fe deficiency response regulators. (**a**) response to P_i_ deficiency in the presence of Fe (−Pi+Fe vs. +P_i_+Fe); (**b**) Fe deficiency response in the presence of P_i_ (+P_i_−Fe vs. +P_i_+Fe); (**c**) response to combined Pi and Fe deficiency compared to P_i_ deficiency (−Pi−Fe vs. −Pi+Fe). After an initial growth period of five days on nutrient sufficient conditions (+P_i_+Fe), seedlings were transferred to the indicated conditions and allowed to grow for an additional six days before harvest. +P_i_: 500 µM, −Pi: 5 µM, +Fe: 50 µM, −Fe: no Fe added. The log_10_ fold changes are color coded. For Col-0, only changes at *p* ≤ 0.05 (Student’s *t*-test two tailed, equal variances) are color coded. For the mutants, changes are color coded, if the metabolic response was different compared to the Col-0 response at *p* ≤ 0.05 (two way ANOVA, *n* ≥ 8). Grey boxes indicate no change in metabolite concentrations in Col-0 by the respective treatment, or no difference in the metabolic response to the respective treatments between mutants and Col-0. Orn: ornithine; Citru: citrulline; Fum: fumarate; Mal: malate; Aco: aconitate; 2OG: 2-oxoglutarate; Suc: succinate; Cit: citrate. Heat maps have been generated according to the data shown in Appendix A.

## Data Availability

We do not report any additional data beyond those shown in this report.

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
