# Peer review of "Modulation of Phosphate Deficiency-Induced Metabolic Changes by Iron Availability in Arabidopsis thaliana"

_ijms, 2021, doi:10.3390/ijms22147609_

Round 1

Reviewer 1 Report

The paper presents the investigations of shoot and root growth as well as metabolic changes in roots and shoots of Arabidopsis thaliana plants subjected to individual and combined Pi or Fe deficiencies.

The paper contributes to the development of knowledge in the field of Plant Nutrition: Physiological and Metabolic Responses, Molecular Mechanisms and Chromatin Modifications.

The paper is well documented, structured and the study is well presented, as a result, I consider and recommend that the paper be accepted for publication.

Author Response

We are very grateful for the very positive comments about our manuscript. Since reviewer 1 did not ask for any changes to the manuscript, all changes, which we performed are listed in the response to the comment of reviewer 2.

Reviewer 2 Report

This is an extremely well written and carefully performed study looking at the interaction of phosphate and iron status on plant metabolism and its molecular regulation.

Methodology is well described and appropriate. Statistical analysis is sophisticated and relevant. Conclusions are supported by the data presented.

MAJOR:

No major comments.

MINOR:

  1. Phosphate deficiency induced by treatment with intravenous iron replacement in iron deficient patients is an active topic in clinical medicine at the moment. It would be helpful to revise the title to indicate that you are speaking about iron deficiency and phosphate deficiency in plants.

2. When the authors spell out the acronyms for the genes they are studying, they mostly capitalize the words in their entirety (although not always). I do not believe this is the standard convention, and the authors should review this and correct if necessary.

3. On line 394, there are some minor grammatical errors. It should read "… this mutant (or possibly "these mutants") also exhibited the most (or "the most striking") metabolic changes…"[If I have misinterpreted the meaning of the sentence, it confirms that it was not written in a clear manner]

Author Response

First of all, we would like to thank the reviewer for the positive comments about our manuscript. The responses to the minor comments are outlined below.

Point 1: Phosphate deficiency induced by treatment with intravenous iron replacement in iron deficient patients is an active topic in clinical medicine at the moment. It would be helpful to revise the title to indicate that you are speaking about iron deficiency and phosphate deficiency in plants.

Response 1: We added the species name to the title, so that it becomes clear, that the study deals with plants.

Point 2: When the authors spell out the acronyms for the genes they are studying, they mostly capitalize the words in their entirety (although not always). I do not believe this is the standard convention, and the authors should review this and correct if necessary.

Response 2: Considering the abbreviations for the genes, we stick to the conventions (uppercase: proteins, lowercase: gene, lowercase italic: mutant. We rechecked the MS and hope that we did not overlook some abbreviations which do not comply with the convention.

Point 3: On line 394, there are some minor grammatical errors. It should read "… this mutant (or possibly "these mutants") also exhibited the most (or "the most striking") metabolic changes…"[If I have misinterpreted the meaning of the sentence, it confirms that it was not written in a clear manner]

Response 3: There was a grammatical error in this sentence, which made it difficult to understand. We corrected it so that it reads now: "Of all analyzed Fe deficiency response mutants, this mutant also exhibited most striking metabolic changes compared to WT under conditions of Pi deficiency. "